# SpecFuse: Ensembling Large Language Models via Next-Segment Prediction

## Abstract

Ensembles of generative large language models (LLMs) can integrate the strengths of different LLMs to compensate for the limitations of individual models. However, recent work has focused on training an additional fusion model to combine complete responses from multiple LLMs, failing to tap into their collaborative potential to generate higher-quality responses. Moreover, as the additional fusion model is trained on a specialized dataset, these methods struggle with generalizing to open-domain queries from online users. In this paper, we propose SpecFuse, a novel ensemble framework that outputs the fused result by iteratively producing the next segment through collaboration among LLMs. This is achieved through cyclic execution of its inference and verification components. In each round, the inference component invokes each base LLM to generate candidate segments in parallel, and the verify component calls these LLMs again to predict the ranking of the segments. The top-ranked segment is then broadcast to all LLMs, encouraging them to generate higher-quality segments in the next round. This approach also allows the base LLMs to be plug-and-play, without any training or adaptation, avoiding generalization limitations. Furthermore, to conserve computational resources, we propose a model exit mechanism that dynamically excludes models exhibiting poor performance in previous rounds during each query response. In this way, it effectively reduces the number of model calls while maintaining overall performance. We conduct extensive experiments using ensembles of five LLMs with different architectures across six benchmarks, covering instruction-response, reasoning, commonsense, and instruction-following tasks. The experimental results demonstrate that SpecFuse consistently enhances performance across all benchmarks, with RougeL scores improving by $+3.1$ on the Chinese and $+3.0$ on the English human-computer interaction benchmarks. Furthermore, the model exit mechanism reduces the average models invoked per round from 5 to 2.4, with only a slight reduction in performance. We will release the code for SpecFuse.

## 1 Introduction

Generative large language models (LLMs) (Brown et al., 2020; Yang et al., 2024) have been widely applied attributed to their impressive performance across various domains, providing efficient support for a broad range of user needs. These off-the-shelf generative LLMs specialize in different areas due to differences in training data and model architecture. Therefore, by combining their strengths, an ensemble of LLMs (Yang et al., 2023) can alleviate the biases and errors of individual models, delivering a better user experience. Unfortunately, vocabulary discrepancies across different LLMs limit the application of traditional logits-based fusion methods (Schapire & Freund, 2013; Sagi & Rokach, 2018) in the integration of generative LLMs.

Recent research on ensembling generative LLMs can be divided into two categories: post-hoc ensemble methods and pre-selection ensemble methods. The post-hoc ensemble method (Jiang et al., 2023b; Lv et al., 2024b) first generates complete responses for a given question by employing all base LLMs, then integrates these responses through a trained fusion model. The pre-selection ensemble method (Lu et al., 2023) pre-trains a query routing model that, for a given query, assigns it to the LLM most likely to generate a high-quality response, using only that LLM for inference.

However, both methods overlook the potential for LLMs to collaboratively generate higher-quality responses through mutual inspiration during the inference process. Additionally, as these methods train an extra fusion model or routing model on specific datasets, they tend to struggle with poor generalization when faced with open-domain queries from users.

In this paper, we introduce SpecFuse, a novel ensemble framework that leverages mutual inspiration between LLMs to produce high-quality next segment. Inspired by Speculative Decoding (Leviathan et al., 2023), SpecFuse achieves this by iteratively executing its two main components: Inference and Verification. In the inference component, given the preceding context, all base LLMs generate candidate fragments simultaneously, with a predefined maximum length per round. The verification component concatenates each newly generated candidate segment with the preceding context to form a batch, then feeds it into each LLM to rank the segments by calculating sequence probabilities in parallel. The top-ranked segment is then broadcast to all LLMs, inspiring them to generate higher-quality segments in the next round. In this process, there is no need to train additional fusion or routing models, which avoids generalization limitations and allows base LLMs to be effortlessly plugged in without any adaptation. Furthermore, to reduce computational costs, we propose the Model Exit mechanism, which dynamically adjusts the softmax temperature based on previous candidate rankings, modifying the distribution of cumulative model scores. Models with scores below a certain threshold will exit the response of current query, freeing up resources for other queries and reducing overall machine deployment.

We select five high-performing models with 7-9 billion parameters as base LLMs and evaluate our framework across six benchmarks, covering instruction-response, reasoning, commonsense, and instruction-following tasks. Experimental results show that SpecFuse consistently enhances performance across all benchmarks, with average Rouge (n) scores improving by $+3.1$ on the English human-computer interaction benchmarks. Furthermore, the model exit mechanism reduces the average number of models invoked per round from 5 to 2.4, with only a slight impact on performance.

In summary, our contributions are as follows:

- We propose SpecFuse, a novel ensemble framework that generates fused results by iteratively producing the next segment through collaboration among LLMs. Our framework allows base LLMs to be effortlessly plugged in without any training or adaptation, thus avoiding generalization limitations.

- We introduce a model exit mechanism that dynamically excludes models with poor performance in previous rounds during each query response, maintaining ensemble performance while reducing computational costs.

- We evaluate our framework on four tasks, including instruction-response and commonsense, and the results demonstrate that SpecFuse consistently enhances performance across six benchmarks. Additionally, the model exit mechanism reduces the average number of models invoked per round by 50%, with only minimal performance loss.

## 2 METHODOLOGY

In the following sections, we first introduce the overall framework of SpecFuse, followed by a detailed explanation of its three parts: the Inference component, the Verify component, and the Model Exit mechanism.

### 2.1 OVERVIEW

Figure 1 shows an overview of SpecFuse. Given $K$ base LLMs $\mathcal{M} = \{m_i\}_{i=1}^{K}$ and an input $I$, for each round in the generation process, SpecFuse first invokes the Inference component (§ 2.2), where the base LLMs in $\mathcal{M}$ generate candidate segments in parallel. Then, it calls the Verify component (§ 2.3) to score each candidate segment, selecting the highest-scoring one as the current round output. Simultaneously, this output is concatenated with the previous input to form the new input $I$ for the next round. Finally, SpecFuse activates a Model Exit mechanism (§ 2.4), removing models with low cumulative scores from $\mathcal{M}$. The above three operations are repeated in a loop until the generated segment contains an end token.

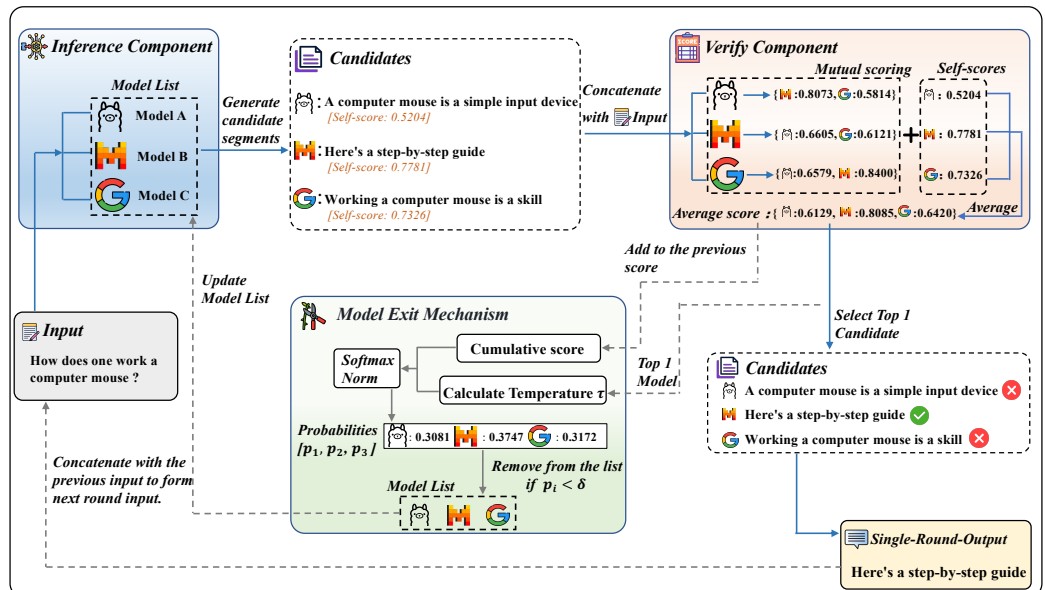

Figure 1: An overview of SpecFuse, a novel ensemble framework, consisting of three parts: the Inference component, the Verify component, and the Model Exit mechanism. The blue solid line represents a single round of the process, while the dashed line shows the process of updating the models participating in the ensemble and refreshing the Input for the next round. In SpecFuse, the Inference component and Verify component synchronously update the model list after the Model Exit mechanism is executed. $\delta$ is the threshold, and when the probability drops below it, the model is excluded from the current generation process.

## 2.2 INFERENCE COMPONENT

Given a maximum length $L$ for the candidate segments generated at each round, the Inference component parallelly invokes each model in $\mathcal{M}$ to generate candidate segments $\{\mathcal{C}_i\}_{i=1}^{|\mathcal{M}|}$, extending from input $I$, and with a length not exceeding $L$, where $|\mathcal{M}|$ denotes the number of models in $\mathcal{M}$. The probabilities corresponding to each token in the candidate $\mathcal{C}_i$, are averaged to produce the self-score $\mathcal{S}_i^i$ of the model $m_i$ :

$$\mathcal{S}_i^i = \frac{\sum_{n=1}^{\hat{L}_i}(x_i^n)}{\hat{L}_i} \tag{1}$$

where $\hat{L}_i$ represents the actual length of candidate $\mathcal{C}_i$, as the generated candidates may be shorter than $L$ in the final round. $x_i^n$ is the probability obtained by applying Softmax normalization to the logits output by model $m_i$ when generating the $n$-th token in $\mathcal{C}_i$. Finally, each model's generated candidate segment, together with its corresponding self-score, is input into the Verify component. We include the model's self-score, as the do-sample[1] method used by generative LLMs involves high randomness and may not yield the model's best segment. If the model scores other candidates higher than its own output, it indicates that its generated text is of lower quality.

## 2.3 VERIFY COMPONENT

The Verify component first concatenates each candidate segment $\mathcal{C}_i$ with the Input $I$, forming the concatenated text $\bar{\mathcal{C}}_i$. These concatenated texts $\{\bar{\mathcal{C}}_i\}_{i=1}^{|\mathcal{M}|}$ are then grouped into a batch, with each model's own generated candidate being removed from the batch to reduce computational load. Next, the Verify component enables all models to compute the probability of each token in the input text in parallel. Similar to obtaining $\mathcal{S}_i^i$, the probabilities of each token in segment $\mathcal{C}_i$ predicted by model $m_j$ are averaged to compute the sequence score $\mathcal{S}_i^j$, representing the evaluation of $\mathcal{C}_i$ by model $m_j$.

---

[1]https://huggingface.co/docs/transformers/generation_strategies

For each candidate $C_i$, its self-score and the scores from other models are averaged to obtain its quality score, denoted as $\tilde{\mathcal{S}}_i$:

$$\tilde{\mathcal{S}}_i = \frac{\sum_{j=1}^{|\mathcal{M}|} \mathcal{S}_i^j}{|\mathcal{M}|} \tag{2}$$

Finally, the candidate segment with the highest quality score is selected as the output presented to users for the current round.

In the implementation process, we use key-values cache to reduce redundant computations of previous text in both the verification and inference stages, improving the inference speed in each round. Collaborating between the inference and verification components to generate the next segment not only alleviates low-quality responses caused by a single model's unfamiliarity with the user's question but also reduces instability from sampling during generation. Furthermore, incorporating the best candidate segment as input for the next round can stimulate other models, and throughout the multi-round generation process, models can continuously inspire one another, ultimately leading to higher-quality responses.

## 2.4 Model Exit Mechanism

While model ensembles can provide users with more stable and higher-quality responses, they also come with increased computational resource demands and costs. To reduce the computational overhead without compromising performance, we propose a Model Exit mechanism. The motivation for this approach stems from our observation that, when responding to a query, some models' output segments rarely rank first. This indicates that these models are not well-suited for responding to the given query, making further computational investment in them inefficient. We use the cumulative scores from previous rounds of each model as prior estimates of quality in subsequent rounds to determine whether a model should be exited. Since the number of rounds varies for different queries, a fixed threshold cannot be used for exit decisions. Therefore, we apply the softmax function to normalize the scores and set a temperature coefficient of $\sqrt{T}$ ($T$ being the current round). We choose $\sqrt{T}$ as the temperature coefficient because the number of output rounds rarely exceeds 100. By using $\sqrt{T}$, we effectively limit the cumulative scores to under 10, preventing extreme values from dominating. Additionally, we analyze the distribution of the best candidate segments. When these segments belong to only a few models, other models can be exited. By combining the temperature coefficient with the best segment distribution, the softmax scores more accurately reflect model performance, allowing underperforming models to exit promptly.

Specifically, we use $Q_i = \sum_{t=1}^{T} \tilde{\mathcal{S}}_i^t$ to represent the cumulative quality score of the candidate segments generated by model $m_i$ from the first step to the current $T$-th step. Next, we count the number of times each model ranked first in previous steps and weighted them based on the step intervals from the current step: within 4 steps by 1, 4 to 8 steps by $3/4$, 8 to 12 steps by $1/2$, and beyond 12 steps by $1/4$, resulting in a weighted count $r_i$ for each model $m_i$. We introduced positional weights when calculating $r_i$ to prioritize recent steps, since models that perform well only in distant steps are less relevant to future responses. The $r_i$ is then normalized to create a distribution $P_r = \{\tilde{r}_i\}_{i=1}^{|\mathcal{M}|}$, where $\tilde{r}_i$ is calculated as follows:

$$\tilde{r}_i = \frac{r_i}{\sum_{j=1}^{|\mathcal{M}|} r_j} \tag{3}$$

Next, we use entropy to measure the uncertainty of the distribution, where for an $n$-model distribution, the entropy ranges from $[0, \log n]$. To facilitate further processing, we normalize the entropy by dividing it by $\log n$, resulting in a value range of $[0, 1]$, and obtain $\mathcal{H}$:

$$\mathcal{H} = \frac{-\sum_{i=1}^{n} \tilde{r}_i \log \tilde{r}_i}{\log n} \tag{4}$$

We can see that lower entropy $\mathcal{H}$ indicates less uncertainty in the distribution $P_r$, where $\tilde{r}$ values are large for a few models and nearly zero for others, suggesting that some models may be discarded. Subsequently, we normalize the cumulative quality score $Q$ using the Softmax function and adjust the temperature coefficient based on $\mathcal{H}$ to control output sharpness, combining both factors to evaluate the likelihood $p$ of a model generating the best candidates in future steps:

$$p_i = \frac{exp(Q_i/\max(1, (\mathcal{H} \times \sqrt{T})))}{\sum_j exp(Q_j/\max(1, (\mathcal{H} \times \sqrt{T})))} \tag{5}$$

The reason for using $\max(1, \cdot)$ is to prevent overly sharp distributions in the early steps of inference when $\mathcal{H} \times \sqrt{T}$ is less than 1, which could lead to mistakenly discarding some models. Finally, we set the threshold value as:

$$\delta = \lambda \times \frac{1}{n} \tag{6}$$

where $\lambda$ is an coefficient and based on experiments on the validation set, we use $\lambda$ as $0.5$. If $p_i$ is less than $\delta$, it will be removed from $\mathcal{M}$ and excluded from the current query response.

# 3 EXPERIMENTS

## 3.1 SETUPS

**Evaluation datasets.**    We evaluate all the models on six datasets that represent different core capabilities of LLMs, open-domain instruction-response (IR), commonsense, reasoning and instruction following.

- Open-domain IR:   We evaluate the model's open-domain instruction-response capability using both English and Chinese human-computer interaction datasets. For the English dataset, we choose the Alpaca-gpt4 (Peng et al., 2023) and Dolly-15k (Conover et al., 2023) datasets for evaluation, both of which have inputs that consist of human instructions. For the Chinese dataset, we utilize the Human-Value and Ruozb datasets from the COIG-CQIA (Bai et al., 2024) benchmark for testing. A detailed description of the dataset can be found in Appendix A.

- Commonsense:   We use the MMLU (Hendrycks et al., 2021), which covers 57 subjects across STEM, and the ARC-C (Clark et al., 2018), which includes questions from science exams for grades 3 to 9, to assess the model's commonsense abilities.

- Reasoning:   To evaluate the model's reasoning abilities, GSM8K (Cobbe et al., 2021) a dataset of high-quality, linguistically diverse grade school math word problems is used.

- Instruction following:   To evaluate the model's instruction-following capability, we utilize IFEval (Zhou et al., 2023), a method specifically designed to assess how proficiently language models follow instructions.

**Base LLMs.**    In our experiment, we chose the top-performing open-source models with parameter sizes ranging from 7 to 9 billion as the base LLMs for our ensemble framework including Llama-3-8B (AI@Meta, 2024), Mistral-7B-v0.3 (Jiang et al., 2023a), Qwen2-7B (Yang et al., 2024), Glm-4-9b (GLM et al., 2024), and Gemma-2-9b (Gemma et al., 2024).

**Evaluation metrics.**    We use a variety of metrics for different tasks, following the test scripts from the Openllm leaderboard. To assess the quality of human question-answering, we apply BARTScore (Bart-S) (Yuan et al., 2021), BERTScore (Bert-S) (Zhang et al., 2019), GPT4-Rank (GPT4-R) (OpenAI et al., 2024), BLEU (Papineni et al., 2002), and ROUGE (R-n) (Lin, 2004). For multiple-choice tasks such as MMLU and ARC-C, we select the option with the highest likelihood to calculate accuracy (Acc). For the reasoning dataset GSM8K, we evaluate exact match (EM) accuracy. For IFEVAL, we rely on the evaluation files provided by the dataset creators (Zhou et al., 2023), testing under prompt-strict, instruction-strict, prompt-loose, and instruction-loose conditions. A detailed explanation of the evaluation methods is provided in Appendix B.

**Baselines.**    Since our approach has not undergone any additional training, we selecte several types of untrained baseline models for comparison with our method: (1) Larger LLMs, including Mixtral-8x7B-v0.1 (Jiang et al., 2023a), Qwen2-72B (Yang et al., 2024), and Llama-3-70B (AI@Meta, 2024). (2) PairRank: an English reward model introduced in the LLM-Blender (Jiang et al., 2023b), which compares candidate results generated by different LLMs and selects the best candidate as the ensemble output. (3) Minimum Bayes Risk (MBR) (Freitag et al., 2023): selects the answer with the highest lexical similarity to other candidate answers. In this paper, we use the SimCSE (Gao et al., 2022) model to calculate the similarity between candidate responses. (4) Generation Fusion (GF) (Jiang et al., 2023b): uses the outputs of other models as context, passing them to a new model, which generates a response based on this context. (5) Majority Voting: each model provides a choice, and the final result is determined by the option with the most votes.

**Implement details.** As the methods in this paper are not trained, we only provide the parameter settings for inference. All models in this study are loaded with bfloat16 precision for inference and use the following generation parameters: $do\_sample = True$, $temperature = 0.6$, and $top\_p = 0.9$. All of our experiments are conducted on A100 GPUs, and set the maximum length of candidate segments to 10.

## 3.2 MAIN RESULTS

| Model | Rouge1↑ | Rouge2↑ | RougeL↑ | BLEU↑ | Bart-S↑ | Bert-S↑ | GPT4-R↓ |
|---|---|---|---|---|---|---|---|
| *Base LLMs* | | | | | | | |
| Llama-3-8B | 25.1622 | 9.7688 | 23.3102 | 3.5669 | -2.9837 | 69.9865 | 9.0850 |
| Glm-4-9B | 25.8456 | 10.2618 | 23.8950 | 3.4774 | -2.9608 | 70.5125 | 8.7993 |
| Qwen2-7B | 26.6179 | 10.8107∗ | 24.4886 | 3.8603 | -2.9380 | 71.4384 | 8.1443 |
| Gemma-2-9B | 25.3130 | 10.0080 | 23.5932 | 4.1933 | -2.9282∗ | 71.5234 | 8.6027 |
| Mistral-7B | 27.7450∗ | 10.7520 | 25.5678∗ | 4.8154∗ | -2.9368 | 71.8773∗ | 7.5093∗ |
| *Larger LLMs* | | | | | | | |
| Llama-3-70B | 26.7744 | 10.8736 | 24.5639 | 4.0981 | -2.8376 | 70.9799 | 5.2153 |
| Qwen2-72B | 27.2580 | 11.2312 | 25.1139 | 4.2896 | **-2.7601** | 71.7302 | 4.1950 |
| Mixtral-8x7B | 29.0371 | 12.2504 | 26.7546 | 4.0820 | -2.8131 | 72.1949 | 3.9461 |
| *Ensemble Base LLMs* | | | | | | | |
| GF (Qwen2) | 23.0829 | 8.9201 | 21.2768 | 3.1881 | -2.9513 | 69.7043 | 9.6143 |
| GF (Gemma-2) | 21.8077 | 7.6626 | 20.0847 | 3.0041 | -3.0178 | 68.1968 | 9.7543 |
| GF (Mistral) | 24.9248 | 9.5800 | 22.9664 | 3.9242 | -2.9263 | 70.3801 | 8.2710 |
| MBR | 27.1221 | 10.4025 | 25.3322 | 4.5642 | -2.8912 | 71.6312 | 7.5003 |
| PairRank | 28.2055 | 10.8611 | 25.9361 | 4.9900 | -2.8637 | 72.0871 | 6.7073 |
| **SpecFuse** | 30.6664 | 13.7367 | 28.3507 | 5.2799 | -2.8653 | 72.8354 | 3.8290 |
| **SpecFuse (w/o ET)** | **30.8566** | **14.0015** | **28.5648** | **5.5113** | -2.8801 | **72.8901** | **3.8227** |

Table 1: Performance on the English Open-Domain Instruction-Response benchmark, with the best result for each metric highlighted in bold and an ∗ indicating the highest result among base LLMs. The upward arrow indicates that a higher value for the metric is better, while the downward arrow indicates that a lower value is better. All ensemble methods in the table integrate all base LLMs, with GF (Qwen2) using the outputs of the other base LLMs as context to generate the fused result through Qwen2-7B.

**Open-Domain Instruction-Response Tasks.** We evaluate the performance of our ensemble framework in responding to user queries on both English and Chinese benchmarks and compare it with single LLMs and other ensemble methods. The test results on the English benchmark are shown in Table 1. The experimental results demonstrate that by integrating base LLMs, SpecFuse surpasses all base LLMs and previous ensemble methods across all metrics, with an average improvement of more than 3 points in the Rouge (n) scores, while also achieving the highest overall ranking in the GPT4-Rank metric, which the responses of all models in the table using GPT-4[2]. Compared to large models with over 70B parameters, our method is competitive across most metrics. This suggests that in open-domain scenarios with uneven instruction difficulty, SpecFuse provides more stable output by integrating the advantage of multiple base LLMs, achieving response quality comparable to larger models while significantly lowering deployment costs and complexity. Previous generation fusion methods, such as GF (Glm-4), exhibit poor integration performance without additional training, and in some metrics, their performance is even worse than that of the individual models. Furthermore, as shown in Table 2, the results on the Chinese benchmark are similar to those on the English benchmark, with SpecFuse outperforming previous ensemble methods and base LLMs across all metrics, demonstrating that its effectiveness is not constrained by language and highlighting its strong generalization ability.

---

[2]https://openai.com/index/gpt-4/

| Model | Rouge1↑ | Rouge2↑ | RougeL↑ | BLEU↑ | Bart-S↑ | Bert-S↑ | GPT4-R↓ |
|---|---|---|---|---|---|---|---|
| *Base LLMs* | | | | | | | |
| Gemma-2-9B | 29.1486 | 7.6523 | 18.3456 | 3.3647 | -4.2845∗ | 68.7312 | 8.5471 |
| Qwen2-7B | 29.9296 | 8.0901 | 20.0345 | 3.6181 | -4.3271 | 69.9989 | 6.5122 |
| Mistral-7B | 30.9890∗ | 8.6498 | 20.6568∗ | 4.4205 | -4.4800 | 70.1000 | 6.6230 |
| Glm-4-9B | 30.8761 | 8.7092∗ | 20.4193 | 4.4674∗ | -4.3018 | 70.2452∗ | 5.2068∗ |
| *Larger LLMs* | | | | | | | |
| Llama-3-70B | 27.7816 | 7.0486 | 20.2227 | 4.1399 | -4.5517 | 68.5211 | 7.4415 |
| Qwen2-72B | 31.4356 | 8.9688 | 22.4781 | **4.8838** | -4.3368 | **70.6480** | 3.4485 |
| *Ensemble Base LLMs* | | | | | | | |
| GF (Qwen2) | 28.6936 | 7.8675 | 18.9339 | 3.3169 | -4.4105 | 69.8126 | 8.3508 |
| GF (Mistral) | 30.2933 | 8.1220 | 20.3324 | 3.8817 | -4.5356 | 70.0437 | 7.1736 |
| GF (Glm-4) | 30.2643 | 8.6996 | 20.5103 | 4.2722 | -4.3316 | 70.2348 | 5.5864 |
| MBR | 30.9335 | 8.7132 | 20.6322 | 4.3149 | -4.3060 | 70.2266 | 4.9921 |
| **SpecFuse** | 31.8931 | 9.3475 | 23.5114 | 4.6383 | **-4.2596** | 70.5199 | 3.7077 |
| **SpecFuse(w/o ET)** | **32.3152** | **9.4461** | **23.7639** | 4.7074 | -4.2759 | 70.5662 | **3.4023** |

Table 2: Performance on the Chinese Open-Domain Instruction-Response benchmark.

| Model | MMLU | ARC-C | GSM8K | IFEVEL | |
|---|---|---|---|---|---|
| | (5-shot) | (5-shot) | (3-shot) | prompt-avg | instruct-avg |
| *Base LLMs* | | | | | |
| Qwen2-7B | 68.2310 | 84.7269 | 74.2229 | 41.6985 | 53.8841 |
| Glm-4-9B | 67.1627 | 85.1535 | 71.7968 | 56.0114 | 67.1393 |
| Gemma-2-9B | 71.5079 | 88.1399 | 77.2555 | 61.6408 | 72.2564 |
| *Ensemble Base LLMs* | | | | | |
| Majority-Voting | 71.7850 | 88.3785 | 77.2927 | – | – |
| MBR | – | – | 76.9832 | 54.9618 | 66.2145 |
| **SpecFuse (Qwen2+GLM4)** | 70.7316 | 87.5372 | 75.4359 | 51.1450 | 63.0703 |
| **SpecFuse (Qwen2+Gemma2)** | 72.1837 | 88.3959 | **78.6960** | 56.0114 | 67.3859 |
| **SpecFuse (GLM4+Gemma2)** | 71.8203 | 88.7372 | 75.8150 | **66.8894** | **75.5241** |
| **SpecFuse (All)** | **73.0117** | **89.0784** | 77.4071 | 62.1107 | 71.5573 |

Table 3: Performance on commonsense, reasoning, and instruction-following tasks. For the IFE-VAL task, we average the prompt-strict and prompt-loose results to obtain prompt-avg and apply the same approach to calculate instruction-avg.

**Commonsense, Reasoning, and Instruction-Following Tasks.** To investigate the performance of SpecFuse on commonsense, reasoning, and instruction-following tasks, we select three base LLMs with different task specializations and conduct experiments on four benchmarks. As shown in Table 3, SpecFuse (All) outperforms previous ensemble methods across all four benchmarks and is not constrained by the task format. Additionally, by integrating different combinations of base LLMs with SpecFuse, we observe the following: (1) When the performance gap between base LLMs is not very large, SpecFuse delivers the most significant overall improvement, with SpecFuse (Qwen2+GLM4) achieving gains of $+2.5$ on MMLU and $+2.4$ on ARC-C by leveraging the respective strengths of Qwen2-7B and GLM-4-9B. (2) When the performance gap is large, integrating three LLMs yields more stable results compared to two. For instance, in the IFEVAL benchmark, Qwen2-7B lags 20 points behind Gemma-2-9B on average, and after integration, the performance improves by 15 points compared to Qwen2-7B but decreases by 5 points compared to Gemma-2-9B. Adding GLM-4-9B to the ensemble brings the performance becomes roughly the same as Gemma-2-9B, illustrating that relying on a single strong model in an ensemble system is insufficient. Frequent updates or the addition of new models are necessary, as the rapid evolution of large models quickly makes previous SOTA LLMs obsolete, showcasing the plug-and-play advantage of our framework.

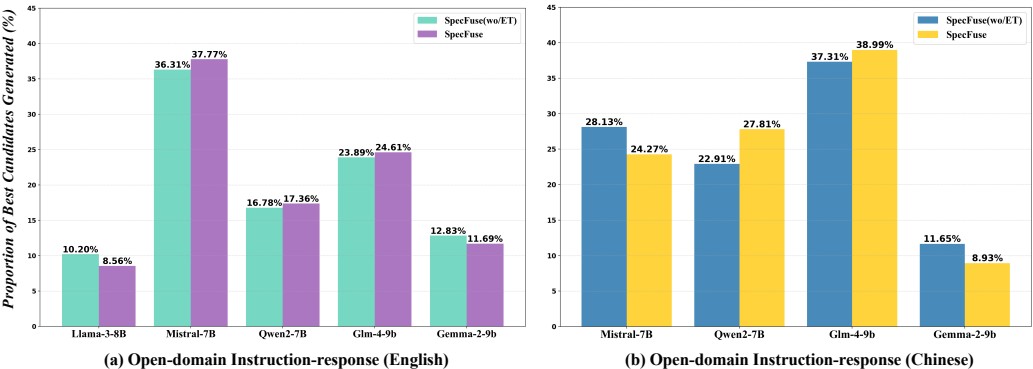

Figure 2: In the test sets of the Open-Domain IR English benchmark and the Chinese benchmark, the percentage of iterations where each model generates the best candidate segment out of the total iterations in the ensemble framework during testing is measured.

## 4 ANALYSIS

In this section, we first conduct ablation studies to analyze the significance of the model exit mechanism in our approach, followed by an analysis of the maximum candidate segment length, the number of base LLMs, and latency. Additionally, the case study is described in Appendix C.

**Ablation On Model Exit Mechanism.** We conduct an ablation study of the Model Exit (ET) mechanism on the Instruction-Response English benchmark test set, using five base LLMs. As shown in Table 4, SpecFuse ($\tau = 1$) results in the fewest model invocations but suffers from significant performance loss. This occurs because, as iterations increase, the cumulative model scores grow larger, causing the softmax function to produce a sharper distribution. Consequently, models with slightly lower scores are prematurely eliminated from the response, which negatively impacts the overall performance of the ensemble. SpecFuse ($\tau = \sqrt{T}$) sets the temperature coefficient to $\sqrt{T}$, which makes the softmax function overly smooth. As a result, it takes many iterations to accumulate substantial score differences between models before the lower-scoring models exit the current response, leading to delayed exits and an excessive number of model invocations overall. The ET mechanism, with dynamic temperature scaling, adjusts the temperature coefficient based on the distribution of the best candidate from previous rounds, ensuring timely model exits. As shown in Figure 2, the proportion of best candidate generations for each LLM changes only slightly with or without ET, indicating that the mechanism primarily eliminates LLMs with low selection probability, minimizing its impact on overall performance while reducing the number of base LLMs invocations.

| Model | Bert-S | BLEU | RougeL | AMIR |
|---|---|---|---|---|
| SpecFuse (w/o ET) | 72.8901 | 5.5113 | 28.5648 | 5.0000 |
| SpecFuse ($\tau = 1$) | 71.7362 | 4.2908 | 25.1196 | 2.0268 |
| SpecFuse ($\tau = \sqrt{T}$) | 72.8105 | 5.3863 | 28.2746 | 4.1996 |
| SpecFuse | 72.8354 | 5.2799 | 28.3507 | 2.4168 |

Table 4: Ablation study of the Model Exit mechanism. SpecFuse ($\tau = 1$) indicates that the Model Exit Mechanism is used but without dynamic temperature scaling, with the temperature fixed at 1, and ($\tau = \sqrt{T}$) indicates the temperature is fixed at $\sqrt{T}$, where T refers to the current iteration of SpecFuse. AMIR refers to the average number of models invoked per iteration.

**Analysis of the Maximum Length of Candidate Segments.** To explore the impact of different maximum generation lengths of candidate segments on the performance of the SpecFuse framework, we conducted tests on the English open-domain IR development set. As shown in Figure 3, the BertScore initially rises with increasing maximum length, reaching its highest point at a length

of 10, after which it begins to decline. When the candidate segment length is too short, it contains insufficient information, affecting the verification component's judgment and making it difficult for models to effectively inspire one another. On the other hand, if the length is too long, the reduced frequency of cross-model interaction leads to less effective knowledge fusion, ultimately diminishing the quality of the final output. Additionally, we observe that as the candidate segment length increases, the first-token latency also increases, but since the total output length does not change significantly, the overall number of system iterations decreases, resulting in more tokens generated per second on average. Based on these observations, we conclude that setting the maximum candidate segment length to 10 during model inference provides the best generation quality, while starting with a shorter length allows users to receive quicker feedback.

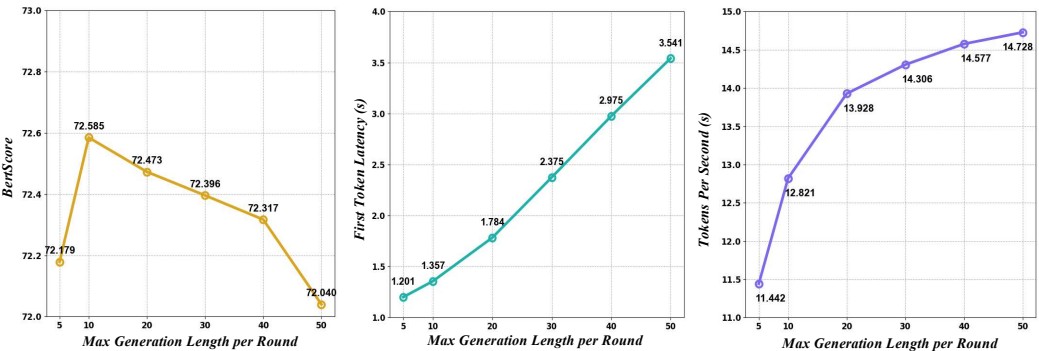

Figure 3: The variation trends of SpecFuse's BertScore, first-token latency, and tokens generated per second as the maximum generation length of each candidate segment changes.

**Analysis of the Number of Base LLMs.**  We test the variation in SpecFuse's performance as the number of base LLMs increases on the test set of the Open-Domain Instruction-Response English benchmark. As shown in Figure 4, the performance of SpecFuse consistently improves as the number of integrated base LLMs increases. When stronger models are introduced, the improvements are substantial, while adding weaker models results in moderate enhancements. This reveals that even weaker models contribute to overall system performance by integrating their strengths, highlighting the advantage of our framework in seamlessly incorporating new LLMs without the need for any training or adaptation.

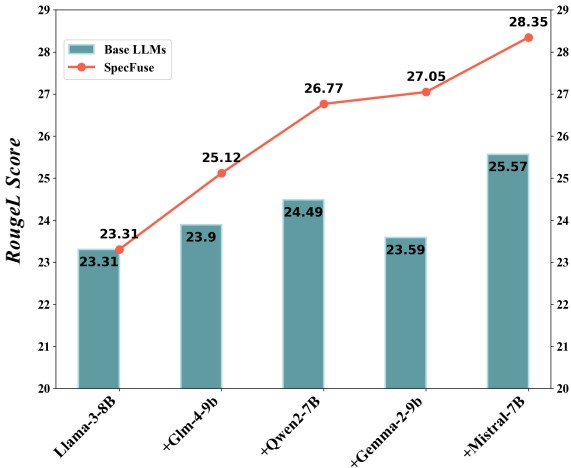

Figure 4: The variation in SpecFuse's RougeL score as the number of base LLMs increases.

| Model | FTL/s | PTL/s |
|---|---|---|
| Llama-3-8B | 0.3491 | 0.0231 |
| Glm-4-9B | 0.5500 | 0.0362 |
| Qwen2-7B | 0.3551 | 0.0258 |
| Gemma-2-9B | 0.3872 | 0.0497 |
| Mistral-7B | 0.3720 | 0.0278 |
| GF (Qwen2) | 5.3347 | 0.0801 |
| GF (Gemma-2) | 4.0994 | 0.0924 |
| GF (Mistral) | 5.3890 | 0.0824 |
| MBR | 7.1975 | 0.0720 |
| SpecFuse | 1.2883 | 0.0765 |

Table 5: Results of the inference latency comparison experiment. FTL refers to first-token latency, and PTL refers to the per token latency. The generated length is fixed at 100 tokens.

**Analysis of Latency.**  Since the time users wait for the system to generate the first token (FTL), as well as the average time per token (PTL), significantly affects the user experience, we conduct

experiments on the Open-Domain IR English benchmark to compare the first-token latency and the average latency per generated token with other methods. We select 200 queries from the development set and instruct the models to generate responses of 100 tokens each. The maximum length of the candidate segments is set to 10. We can see from Table 5 that in previous ensemble methods, the first-token latency remains around 5 seconds for a response limited to 100 tokens, and this time doubles as the response length increases to 200 tokens, significantly reducing the user experience due to the prolonged waiting time. Compared to them, SpecFuse reduces the first-token latency by 3 to 6 times and can further decrease it by adjusting the maximum length of the candidate segments generated in the first round. For PTL, SpecFuse also consumes less time compared to generation fusion methods, generating more tokens per second. Additionally, since it takes approximately 0.2 seconds for a human to read each token, these models can meet the reading requirements of users. However, the acceptable FTL for users is typically 1-2 seconds, and previous methods frequently encounter the issue of excessively high FTL during online user interactions. In contrast, our model meets this requirement while delivering better ensemble performance.

## 5 RELATED WORK

**Post-hoc ensemble.**   The basic idea of the post-hoc ensemble (Jiang et al., 2023b; Lv et al., 2024a) method is that all base large models first generate complete responses independently for a given question, and then these responses are fused to produce the final ensemble response. Jiang et al. (2023b) trained a reward model, PairRank, to compare pairs of candidate results generated by multiple LLMs, selecting the highest-quality candidate as the ensemble output. As these selection-based methods may restrict the ability to fully utilize each candidate's strengths, Jiang et al. (2023b) proposed LLM-Blender, which first ranks the candidates using PairRank and then uses a trained fusion model to take the top few candidates as context to generate a fused output. Subsequently, Lv et al. (2024b) proposed the URG ensemble method, an end-to-end framework that first ranks responses, selects the top few candidates as context, and then generates a new response.

**Pre-selection ensemble.**   The pre-selection ensemble methods (Lu et al., 2023; Wang et al., 2024; Shnitzer et al., 2023) test base models on a dataset, maps each question to its best-performing model, and then train a routing model using this mapping data to classify future questions to the most proficient model. Lu et al. (2023) trained a reward model that scores the given query and routes it to the highest-scoring model, inferring from that model alone. Similarly, Wang et al. (2024) trained a model called Frugal using expert LLM outputs on training data, and during inference, it obtains encoded vectors from all LLMs for a given query, which are then fed into Frugal to select one expert LLM that produces the final prediction.

However, these two methods focus on either selecting a single LLM's response or merging the complete responses from LLMs, resulting in overlooking the potential for mutual inspiration among LLMs to collaboratively generate higher-quality responses during inference. Additionally, due to the need for training an additional fusion or routing model on specific datasets, these methods tend to struggle with poor generalization when handling open-domain queries from users. In contrast, our SpecFuse framework generates fused results by iteratively producing the next segment through collaboration among LLMs, which supports easy plug-and-play integration of LLMs without requiring any training, thereby avoiding generalization issues.

## 6 CONCLUSION

In this paper, we introduce SpecFuse, a novel ensemble framework that generates fused outputs by iteratively producing the next segment through collaboration among LLMs, allowing base LLMs to be seamlessly integrated without any training or adaptation. Additionally, SpecFuse employs a model exit mechanism that dynamically excludes underperforming models in previous rounds during query responses, reducing computational costs. Experimental results across six benchmarks demonstrate that SpecFuse consistently delivers more stable performance compared to single LLMs and previous ensemble methods. We hope our work inspires further research on online model ensemble, improving the quality of responses delivered to users based on existing LLMs.

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

## A  Dataset Details

The following provides a detailed description of the evaluation of instruction-response capability using the datasets.

For the English dataset, we choose the Alpaca-gpt4 (Peng et al., 2023) and Dolly-15k (Conover et al., 2023) datasets for evaluation, both of which have inputs that consist of human instructions. We select these two datasets because their response sources differ: the Dolly-15k dataset features human-provided responses, while the Alpaca-GPT-4 dataset contains responses generated by the state-of-the-art GPT-4 (OpenAI et al., 2024) model, which provides neutral answers to each question and can refuse to answer inappropriate or harmful ones. Using both types of responses for scoring allows us to more thoroughly compare the advantages of our ensemble system. Additionally, due to the large size of these datasets, we randomly sample portions from each to create a new test set. From the Dolly-15k dataset, we randomly select 1,500 open-QA samples for testing, with 500 reserved for the development set. In the Alpaca-GPT-4 dataset, after shuffling the data, we manually verify the correctness of GPT-4's responses and select 2,000 validated samples, with 1,500 used for testing and 500 for validation.

For the Chinese dataset, we utilize the Human-Value and Ruozb datasets from the COIG-CQIA (Bai et al., 2024) benchmark for testing. The instructions in these two datasets consist of human-posed questions, with answers provided either by humans or generated by GPT-4. The COIA authors manually review and filter the responses, retaining only the correct answers generated by GPT-4.

## B  Evaluation Methods

To evaluate the quality of our framework's responses to human questions in the dataset, a range of metrics assessing model generation capabilities are selected for the following experiments.

- BLEU (B-$n$) (Papineni et al., 2002) and ROUGE (R-$n$) (Lin, 2004) compare a generated response with a reference by calculating $n$-gram overlap. For the Chinese results, we use Jieba[3] to split the text into words before calculating these two scores.

- BERTScore Zhang et al. (2019) (comprising Precision, Recall, and F1-score) measures the similarity between two texts based on the contextualized embedding from BERT (Devlin et al., 2019). In this paper, we report the F1 score of BERTScore.

- BARTScore (Yuan et al., 2021) is a unified evaluator which evaluates with the average likelihood of the pretrained encoder-decoder model, BART (Lewis et al., 2019). It can predict different scores depending on the formats of the source and target.

- The GPT4-Rank evaluation utilizes the GPT-4[4] model to compare two different responses against a ground-truth response. The model will select the better of the two responses.

---

[3] https://pypi.org/project/jieba/

[4] The version we use is GPT-4-turbo, and the link is https://openai.com/index/gpt-4/

For each test sample, we pair the responses generated by different models and have GPT-4 determine which one is superior. Since the MBR and PairRank methods do not generate new responses, we do not re-rank the responses they select from the base LLMs. Instead, we use the average rankings of the responses they select from the base LLMs to represent their GPT4-Rank. Once all comparisons are complete, we count the number of wins for each model. Based on these win counts, we rank the responses from the different models. The average ranking of each model across all data in the dataset is the value reported in our table. The evaluation instructions for GPT-4 are shown in Table 6.

| | |
|---|---|
| **Template** | Instruction: ${instruction} |
| | Ground-Truth Response: ${Truth response} |
| | Model A: ${A response} |
| | Model B: ${B response} |
| | Given the User's Instruction and Ground-Truth response above, please compare the two Model's responses. You only have 2 choices to output: If you think A is better, please output: 1. A is better If you think B is better, please output: 2. B is better Output your choice below: |
| **Comparison Option** | 1. A is better 2. B is better |

Table 6: The template used for GPT-4 compares two models' responses.

## C  CASE STUDY

Table 5 presents a case from the SpecFuse workflow where the user's request is "Write a simile to describe a person who is hard-working." The reasoning process goes through four iterations, and the Verify model's selection of the best candidate is not always from the same model. In the first round, the best candidate is generated by Qwen2. In the second round, Mistral, after receiving Qwen2's output from the previous round, is inspired and generates a response that better meets the user's needs, as using "farmer" to describe a hard-working person is inappropriate. Additionally, the table shows that through mutual inspiration between models, the final response more closely matches the user's expectations, thereby improving the overall quality of the reply.

| User instruction: Write a simile to describe a person who is hard-working. | | | | |
|---|---|---|---|---|
| Input for each round | Qwen2 Generation | Mistral Generation | Glm-4 Generation | Best Candidate |
| *Round 1:* 
 Write a simile to describe a person who is hard-working. | A person who is hard-working is like a determined | This person is as dedicated to their work as | A hard-working person is like a well | A person who is hard-working is like a determined |
| *Round 2:* 
 Write a simile to describe a person who is hard-working. A person who is hard-working is like a determined | farmer, tirelessly cultivating the land and nurturing their crops | river, steadily carving its path through the | horse pulling a heavy cart through a long journey. | river, steadily carving its path through the |
| *Round 3:* 
 Write a simile to describe a person who is hard-working. A person who is hard-working is like a determined river, steadily carving its path through the | rockiest of landscapes. <\|im_end\|> | mountains. | rocks and obstacles in its way, never giving up | rocks and obstacles in its way, never giving up |
| *Round 4:* 
 Write a simile to describe a person who is hard-working. A person who is hard-working is like a determined river, steadily carving its path through the rocks and obstacles in its way, never giving up | until it reaches the sea. | until it reaches its destination. | . <\|user\|> | until it reaches its destination. |
| Overall Output: A person who is hard-working is like a determined river, steadily carving its path through the rocks and obstacles in its way, never giving up until it reaches its destination. | | | | |

Figure 5: Case study of SpecFuse integrating the base LLMs Qwen2, Mistral, and GLM-4. The Best Candidate is the top-ranked option determined by the verify component and is directly presented to the user. $< |\text{im\_end}| >$, $< /s >$, and $< |\text{user}| >$ are special end tokens for the three base LLMs, and generation halts when the best candidate includes any of these end tokens.

