# OpenReview forum: "SpecFuse: Ensembling Large Language  Models via Next-Segment Prediction"
_ICLR.cc/2025/Conference — Submitted to ICLR 2025_

### Official Review · Reviewer_PsEZ · 2024-10-29

**Soundness:** 2
**Presentation:** 3
**Contribution:** 2
**Rating:** 5
**Confidence:** 4

**Summary:**

The paper proposes a new approach to leveraging multiple LLMs for better performance. The authors divide generation of a response into several rounds (each round corresponding to generation of  a segment). In each round, the output of a model is judged by all the LLMs, and the output with the highest score is selected as the segment for the round. To balance efficacy and efficiency, the authors also propose an exit mechanism, where models with scores lower than a threshold are removed from ensemble. Empirical studies are conducted on a variety of benchmarks, and evaluation results look good to indicate the effectiveness of the proposed approach.

Here are my concerns:
1) As 1/3 contributions, the exit mechanism does not receive enough analysis in empirical studies. The authors only show that the mechanism leads to slight performance drop, and mention that it also significantly reduces the number of models invoked in Abstract and Introduction, but how does this mechanism work? Will awkward models be kicked out as expected for specific queries? How does the number of models vary with respect to different tasks and different queries? Given that there is a lot of heuristics in the mechanism, it is important to clearly reveal how the mechanism works and justify why it should be designed like that. ( Note that I just list a few questions here, it is better to have a comprehensive study regarding to the mechanism).
2) The authors categorize existing work into two groups: post-hoc and pre-selection, but only involve one post-hoc method in comparison. Why there are no pre-selection methods (e.g., based on a certain router) selected? It is also better to involve more post-hoc methods as baselines.
3) I also worry about latency, especially when responses become long (e.g., math with COT) and models become large (e.g., from 7B to 70B). Can you directly compare different methods in terms of time and resources required for a query, and discuss how the metrics vary according to responses in different lengths? I am also confused about the concept "first token latency", can you clearly define it?

------------------------------------------------------
The authors' response addressed some of my concerns.  My judgment is similar to Reviewer Y7i1, my score is between 5 and 6. The authors tried their best to show the advantages of their method, but I suspect if we really need such complicated ensemble procedure  in practice. Moreover, the comparison is limited to instruction-following. A more comprehensive comparison on tasks like GSM8K, MMLU et al., is expected. Anyway, I do not champion the work, but am OK if it is finally accepted.

**Strengths:**

1. A new approach for LLM ensemble.
2. Extensive empirical studies across various benchmarks.

**Weaknesses:**

Please refer to my concerns in Summary.

**Questions:**

In addition to those I mentioned in Summary

1) Can you compare the proposed methods with Best-of-N (an extra reward model might be needed)?

---

### Official Review · Reviewer_5uRM · 2024-11-04

**Soundness:** 2
**Presentation:** 3
**Contribution:** 2
**Rating:** 5
**Confidence:** 3

**Summary:**

This paper introduces an innovative framework called SpecFuse, designed to ensemble LLMs through next-segment prediction. SpecFuse operates by iteratively executing its inference and verification components. In each round, candidate segments generated by base LLMs are ranked, and the top-ranked segment is broadcast to all models to facilitate collaborative generation and enhance response quality. The framework also incorporates a “Model Exit Mechanism” that dynamically excludes underperforming models to optimize computational resources. Experimental results demonstrate that SpecFuse achieves superior performance compared to individual models and traditional ensemble methods across six benchmark datasets,.

**Strengths:**

1. SpecFuse introduces a novel and practical approach for model ensembling by enabling cooperative generation among multiple LLMs, thereby mitigating the low-quality outputs often associated with single-model approaches.
2. Experimental results highlight significant performance improvements across various tasks, including instruction-response, reasoning, and commonsense benchmarks.
3. The methodology is clearly articulated, with diagrams effectively illustrating the operational mechanics of the framework.

**Weaknesses:**

1. SpecFuse relies on accessing logits, which limits its applicability to models that require API-based interactions, thus constraining its usage.
2. The framework demands substantial additional computational resources, but the paper lacks a comprehensive discussion on this aspect. Additionally, it is necessary to evaluate whether the performance gains justify the increased resource consumption.
3. The paper does not provide comparisons with other ensemble inference enhancement methods, which would contextualize the advantages of SpecFuse more effectively.

**Questions:**

1. Regarding Weakness 2: Could you elaborate on the computational trade-offs and whether the observed improvements are justified by the additional resource requirements?
2. Related to Weakness 3: Can you include comparisons with other existing ensemble inference enhancement techniques?
3. Have you considered testing SpecFuse on more representative or real-world datasets to better illustrate its practical effectiveness?

---

> ### Comment · Reviewer_5uRM · 2024-12-01
>
> Thanks for your detailed experiments and responses. I have adjusted my rating.
>
> For Con 2,  1. can you use a more advanced evaluation method, such as LLM-as-a-Judge or Human Evaluation? since RougeL and BertScore may not suitable for LLM-generations. 2. Can you conduct a significance test? 3. are Self-consistency and CoT, ToT, GoT comparable baselines?

---

### Official Review · Reviewer_y7i1 · 2024-11-06

**Soundness:** 2
**Presentation:** 3
**Contribution:** 3
**Rating:** 6
**Confidence:** 5

**Summary:**

This paper introduces SpecFuse, a novel technique for ensembling LLMs by generating multiple segments in parallel and reranking them using the average log probabilities assigned by each model. The paper also introduces an early exit mechanism which prunes out LLMs from the ensemble which do not contribute the maximum scoring segments.

Results show that this ensembling technique leads to notable improvements when combining base LLMs in the 7-9B parameter range, often outperforming larger models in the 70B range.

**Strengths:**

- The main idea behind SpecFuse is intuitive and interesting -- and empirical results show its effectiveness.
- SpecFuse does not require training any additional modules and can be plugged into any set of existing models.
- Several datasets are used for evaluation and there seems to be a consistent gain across all of them.

**Weaknesses:**

- A very similar technique called Mixture of Agents [1] was introduced recently, which the paper does not mention. The key difference is that this paper divides the response into chunks before aggregating, whereas MoA does it at the response level. Nevertheless, a comparison between the methods would be useful.
- The main evaluation is done on the Alpaca-GPT4 datasets -- where the ground truth references come from GPT-4 responses. This is a curious choice -- the standard way of evaluating instruction following in models is via AlpacaEval, where a separate LLM evaluator is used to grade the responses. Such an evaluation would help situate the proposed method with other techniques proposed in the literature (e.g., MoA above).
- The model exit mechanism seems rather ad-hoc. It would be useful to provide an ablation study showing that the different components (e.g., cumulative quality score, entropy) are indeed needed. A comparison to simpler baselines (e.g., remove all models which do not contribute in the first n segments) would also help.

[1] Wang, Junlin, et al. "Mixture-of-Agents Enhances Large Language Model Capabilities." arXiv preprint arXiv:2406.04692 (2024).

**Questions:**

- For the MMLU and ARC-C datasets, the answer is one of the provided options -- what is the sequence being decoded in these cases? How long are the generations?

---

### Meta-Review · Area_Chair_eTUk · 2024-12-22

**Metareview:**

This paper proposes a training-free method for ensembling multiple language models together. The method works by generating candidate segments and then using all models to rerank them based on average log probabilities. The authors also proposed an exit mechanism to discard models with low scores to reduce computational costs. Experiments on Alpaca-GPT4 demonstrate the effectiveness of the proposed approach.

Strengths:
1. The method is simple and training-free yet works well in practice.

Weaknesses:
1. A major concern raised by reviewers is whether the increased computational cost is necessary and practical in real applications.
2. Some reviewers are also concerned about the generality of the proposed method on tasks other than instruction following.

Overall, this is a very simple and effective method. However, overall none of the reviewers seems enthusiastic about accepting this paper. Also, there seem to be many additional experiments during the rebuttal period and I think incorporating them into the paper can significantly improve it for the next version. Therefore, I'm recommend rejection, but I wouldn't mind if the paper gets accepted.

**Additional Comments On Reviewer Discussion:**

In addition to the points above, reviewers pointed out comparisons with existing works such as Mixture of Agents, which the authors have addressed with additional experiments. Reviewers also raised concerns about Alpaca-GPT4 used instead of the more common AlpacaEval, which the authors also addressed. Reviewers also raised concerns about ablation studies of the exit mechanism, which the authors have also addressed during rebuttal.

---

### Decision · Program_Chairs · 2025-01-22

Reject